# Effects of unsaturated fatty acids (Arachidonic/Oleic Acids) on stability and structural properties of Calprotectin using molecular docking and molecular dynamics simulation approach

Nematollah Gheibi[1], Mohamad Ghorbani[2], Hanifeh Shariatifar[3], Alireza Farasat[1] *

1 Cellular and Molecular Research Center, Research Institute for Prevention of Non Communicable Diseases, Qazvin University of Medical Sciences, Qazvin, Iran, 2 Department of Nanobiotechnology/ Biophysics, Faculty of Biological Science, Tarbiat Modares University, Tehran, Iran, 3 Young Researchers and Elite Club, Tehran Medical Sciences, Islamic Azad University, Tehran, Iran

* a.farasat@qums.ac.ir

**Data Availability Statement:** All relevant data are within the paper and its Supporting Information files.

## Abstract

Calprotectin is a heterodimeric protein complex with two subunits called S100A8/A9. The protein has an essential role in inflammation process and various human diseases. It has the ability to bind to unsaturated fatty acids including Arachidonic acid, Oleic acid and etc., which could be considered as a major carrier for fatty acids. In this study we aimed to appraise the thermodynamics and structural changes of Calprotectin in presence of Arachidonic acid/Oleic acid) using docking and molecular dynami simulation method. To create the best conformation of Calprotectin-Oleic acid/Arachidonic acid complexes, the docking process was performed. The complexes with the best binding energy were selected as the models for molecular dynamics simulation process. Furthermore, the structural and thermodynamics properties of the complexes were evaluated too. The Root Mean Square Deviation and Root Mean Square Fluctuation results showed that the binding of Arachidonic acid/Oleic acid to Calprotectin can cause the protein structural changes which was confirmed by Define Secondary Structure of Proteins results. Accordingly, the binding free energy results verified that binding of Oleic acid to Calprotectin leads to instability of S100A8/A9 subunits in the protein. Moreover, the electrostatic energy contribution of the complexes (Calprotectin-Oleic acid/Arachidonic acid) was remarkably higher than van der Waals energy. Thus, the outcome of this study confirm that Oleic acid has a stronger interaction with Calprotectin in comparison with Arachidonic acid. Our findings indicated that binding of unsaturated fatty acids to Calprotectin leads to structural changes of the S100A8/A9 subunits which could be beneficial to play a biological role in inflammation process.

## 1. Introduction

The S100A8/A9 or Myeloid Related Proteins (MRP8/ MRP14) are two small proteins which structurally belong to S100 protein family [1, 2]. The S100A8/A9 protein complex

**Funding:** The author(s) received no specific funding for this work.

**Competing interests:** The authors have declared that no competing interests exist.

constitutes a heterodimeric complex which is highly expressed by myeloid lineage leukocytes, macrophages and lower in monocytes [3]. Generally, the complex includes 40–45% of the cytosolic proteins of human neutrophils which is mainly secreted by the activated neutrophils. The heterodimeric complex is also termed Calgranulin A/B or Calprotectin [4]. Commonly, each subunit of the protein consists of two EF-hands with helix-loop-helix structural motif related by a central hinge region. Moreover, each monomer has the ability to bind to two calcium ions and other divalent cations such as: $Zn^{2+}$, $Cu^{2+}$, $Mn^{2+}$ and $Fe^{2+}$ [5, 6]. The human S100A8/A9 complex has the molecular weight of 11 and 13 kDa with 93 and 114 amino acids respectively [7, 8]. Various configurations of the S100A8/A9 complex are available including homodimers, heterodimers and heterotetramers which the heterodimer form has the highest stability and it is the most important agent for protein interactions [5, 9, 10]. Due to absence of signal peptide sequence in the aforementioned complex, the protein secretion is done *via* an energy dependent PKC-activation procedure, not by endoplasmic reticulum-Golgi pathway [4, 11, 12]. Moreover, other cell lines including keratinocytes, chondrocytes and endothelial cells have the ability of Calprotectin expression and secretion under certain conditions [5, 13]. The human S100A8/A9 can be considered as an antimicrobial agent due to the capability of the complex to bind to multiple metal ions. Furthermore, several studies indicated that the heterodimer complex can bind to unsaturated fatty acids (UFAs) including arachidonic acid (AA), so it also could be assumed as an important agent in eicosanoids metabolism respectively [1, 14]. The Calprotectin serum level is increased in inflammation process and has a great role in different human diseases such as: rheumatoid arthritis, cystic fibrosis, acute and chronic inflammatory disorders and cardio vascular disease. The Calprotectin can activate signaling cascades in several human diseases which occurs when the protein binds to its specific receptors as target proteins such as: receptor for advanced glycated end products (RAGE), heparan sulphate proteoglycan and toll like receptor (TLR4) which is the main receptor for binding to heterodimeric complex [4, 5, 15, 16]. The Calprotectin also has the ability to regulate the accumulation of neutrophils and macrophages, cytokine production and various processes such as: leukocyte migration, differentiation of myeloid cells, cytoskeleton rearrangement and fatty acid transport [17, 18]. Several studies revealed that the lipid membrane of all eukaryote cells consists of poly unsaturated fatty acids (PUFAs) including n-3 and n-6 fatty acids which they are categorized dependent upon the last carbon-carbon double bond location from the omega carbon [19]. Furthermore, the UFAs can influence various processes including inflammation which is one of the most important phenomena in several human diseases. Due to poor solubility of fatty acids in biological fluids, presence of certain protein carriers is very essential for their transportation to their site of action. The calprotectin is considered to be the main fatty acid carrier of neutrophils with high binding affinity potential and can bind UFAs such as AA, Linoleic acid (LA) and Oleic acid (OA) in a calcium dependent manner. This binding leads to UFAs movement between neutrophils cytosol and the plasma membrane which may have a role in fatty acid uptake [20]. Several studies have proved that Molecular Dynamics (MD) simulation method has a great role in ligand-receptor recognition in a variety of biological processes. Although some aspects of the protein structural changes were evaluated experimentally [20], nonetheless their whole mechanism of action was not considered thoroughly. Thus, in this study we aimed to evaluate the thermodynamics and conformational changes of Calprotectin subunits (S100A8/A9) in presence of UFAs, (AA/OA) in physiological concentration using MD simulation method. Additionally, in current study, it has been proved that binding UFAs to Calprotectin can change the secondary structure and the stability of the protein. Thus, it could be suggested that these changes can influence the binding of Calprotectin complex to other target proteins which are involved in inflammatory processes.

## 2. Methods

### 2.1. Molecular docking and MD simulation analysis

The x-ray crystal structure of the Calprotectin (entry code: 1XK4) [21], AA (entry code: 4DE6) which is obtained from Horse spleen apo-ferritin [22] and OA (entry code: 2FTB) which is acquired from liver bile-binding protein [23] were provided from RCSB Protein Data Bank (PDB). To estimate the permissible torsions for the ligand and characterize the search space coordinates, the graphical AutoDock tool was applied [24]. Afterwards the docking process was performed *via* grid size of $48 \times 50 \times 58$ along the X, Y, and Z axes with 1 Å spacing. The lowermost binding energy of Calprotectin-AA and Calprotectin-OA complexes was created using AutoDock Vina which the complexes were assumed as the primary conformation for MD simulation process [25]. In this study, the MD simulation process was done using GRO-MACS program version 5.1 and the CHARMM 36 force field was used for all simulations. The Calprotectin, Calprotectin-AA and Calprotectin-OA complexes were solvated by transferable intermolecular potential with 3 points (TIP3P) water model in a cubic box with a distance of 10 Å from the furthest atom of the protein. After solvation, $Na^+$ and $Cl^-$ ions were added to neutralize the system. Then, the concentration of 150 mM NaCl and $CaCl_2$ were inserted to the systems [26, 27] and the energy minimization was done using the steepest descent method. Each system was equilibrated by 1 ns MD simulation in the canonical (NVT) ensemble and 1 ns MD simulation in the isothermal–isobaric (NPT) ensemble using position restraints on the heavy atoms of the protein to allow for the equilibration of the solvent. The Nose–Hoover thermostat constant was utilized for fixing the temperature of the system at 310 K. To maintain the pressure of the system at fixed 1bar pressure, the Parrinello–Rahman pressure coupling method was used [28]. The electrostatic interactions were measured using the Particle Mesh Ewald (PME) method with 1.0 nm short-range electrostatic and van der Waals cutoffs [29, 30]. Finally, for each complex, the MD simulation process was repeated twice for about 100 ns with time steps of 2 fs on equilibrated systems respectively. Furthermore, to study the interactions between fatty acids and Calprotectin, the Root Mean Square Deviation (RMSD), Root Mean Square Fluctuation (RMSF), Define Secondary Structure of Proteins (DSSP), Hydrogen bond (H-bond), electrostatic and van der Waals energies of each system were measured using GRO-MACS software during the simulation. Consequently, the Pymol software was applied to illustrate the final PDB file of MD simulation process. In the final step to evaluate the hydrophobic reactions of the complexes and their H-bonds, the Ligplot software was used [31].

### 2.2. MMPBSA

The molecular mechanics Poisson-Boltzmann surface area (MMPBSA) method is widely used to evaluate the molecular model affinities such as: protein-protein and ligand-protein interactions [32]. In this study the binding free energy between S100A8/A9 subunits in presence and absence of AA and OA were analyzed during equilibrium phase by taking snapshots at an interval of 50 ps from 80–100 ns MD simulation using g_mmpbsa, GROMACS software [33].

## 3. Results and discussion

### 3.1. Molecular docking and MD simulation analysis

In this study, to observe the best binding manner of AA and OA to Calprotectin, the molecular docking was done. The results revealed that AA and OA bind to hydrophobic regions of Calprotectin with the lowest binding energy, -5.8 and -6.3 kcal/mol respectively. Thus, the best structure of the complex with the lowest binding energy was created using AutoDock Vina software, which was considered as a model for MD simulation. After the simulation, in order

to evaluate the complex stability, their RMSD profiles were analyzed during 100 ns simulation process. The RMSD profiles were illustrated in Fig 1.

RMSD is an essential parameter which is applied to predict the system equilibration during the simulation [34–36]. As illustrated in the figure, the Calprotectin-fatty acids complexes and the Calprotectin alone were equilibrated after 8 and 6 ns respectively. Moreover, the RMSD average of the Calprotectin-AA/OA complexes and also the Calprotectin alone for the last 5 ns was 0.21 ± 0.03 nm, 0.23 ± 0.02 nm and 0.43 ±0.02 nm respectively. The results showed that RMSD value of the Calprotectin alone was greater than the Calprotectin–UFAs complexes which confirms the structural changes of the Calprotectin in presence of fatty acids. RMSF value provides a better understanding of the protein flexibility and structural fluctuations [37, 38]. To study the Calprotectin flexibility in presence of AA and OA, the RMSF value was calculated for Calprotectin, Calprotectin-AA and Calprotectin-OA during the simulation. The RMSF value of the aforementioned complexes and the Calprotectin were illustrated in Fig 2. As shown, the Calprotectin has the lowest flexibility in a majority of residues in presence of AA and OA fatty acids.

These results confirm that the aforementioned fatty acids have an interaction with Calprotectin which leads to the lowest fluctuation of the Calprotectin residues in interaction with fatty acids. The Fig 3, illustrates the 2D and 3D images of the Calprotectin in interaction with AA and OA after 100 ns MD simulation. As revealed in 3D figure, the AA and OA bind to hydrophobic region which displayed in red.

The Fig 3A shows that in binding Calprotectin to AA, Phe68, Gln69, Leu72, Ile73 and Ile76 amino acids of A8 subunit and Ala84, Thr87 and Trp88 of A9 subunit are involved. As mentioned before, in A8 subunit all the amino acids except Gln69 have high hydrophobicity while in A9 subunit all three amino acids possess high hydrophobicity. The Fig 3B illustrates that in binding Calprotectin to OA, some amino acids including Ile60, Phe68, Gln69, Leu72 and Ile73 of A8 subunit and Ala84, Thr87 and Trp88 of A9 subunit are involved. The same as above, all amino acids except Gln69 possess high hydrophobicity. Based on these results, in binding Calprotectin to fatty acids, four residues of A8 subunit including Phe68, Gln69, Leu72 and Ile73 with three residues of A9 subunit such as: Ala84, Thr87 and Trp88 are common. In one study,

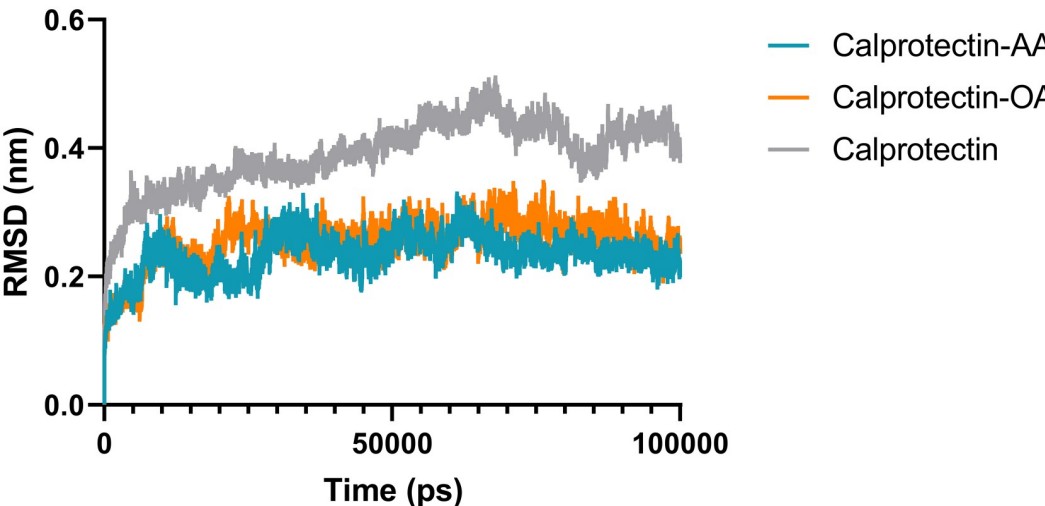

**Fig 1. The root mean square deviation (RMSD) value of Calprotectin Cα alone and the Cα of Calprotectin-AA/ Calprotectin-OA complexes.** As shown in the figure, the RMSD average of the Calprotectin-AA, Calprotectin-OA complexes and the Calprotectin alone for the last 5 ns was 0.21 ± 0.03 nm, 0.23 ± 0.02 nm and 0.43 ±0.02 nm.

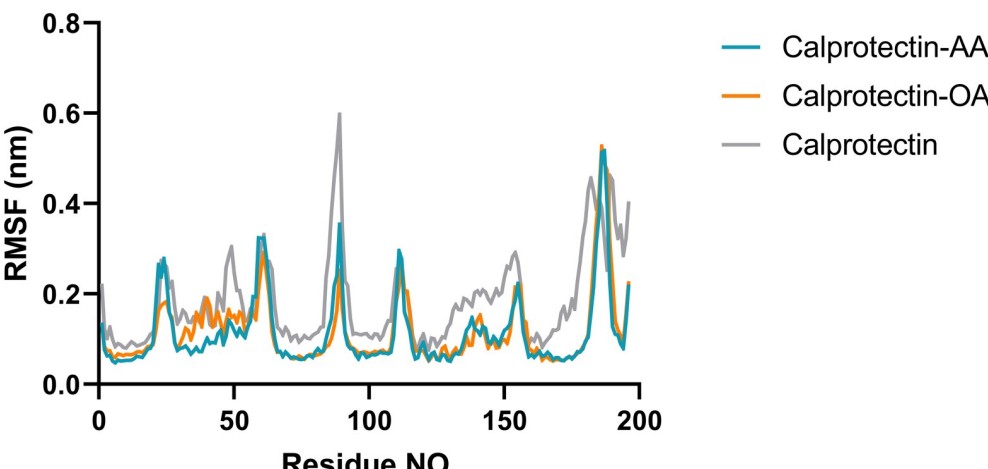

**Fig 2. The RMS fluctuation value of Calprotectin alone and the Calprotectin-AA/ Calprotectin-OA complexes.** As shown, the Calprotectin has the lowest flexibility in a majority of residues in presence of AA and OA. These fatty acids have an interaction with Calprotectin which cause the lowest fluctuation of the Calprotectin residues in interaction with fatty acids.

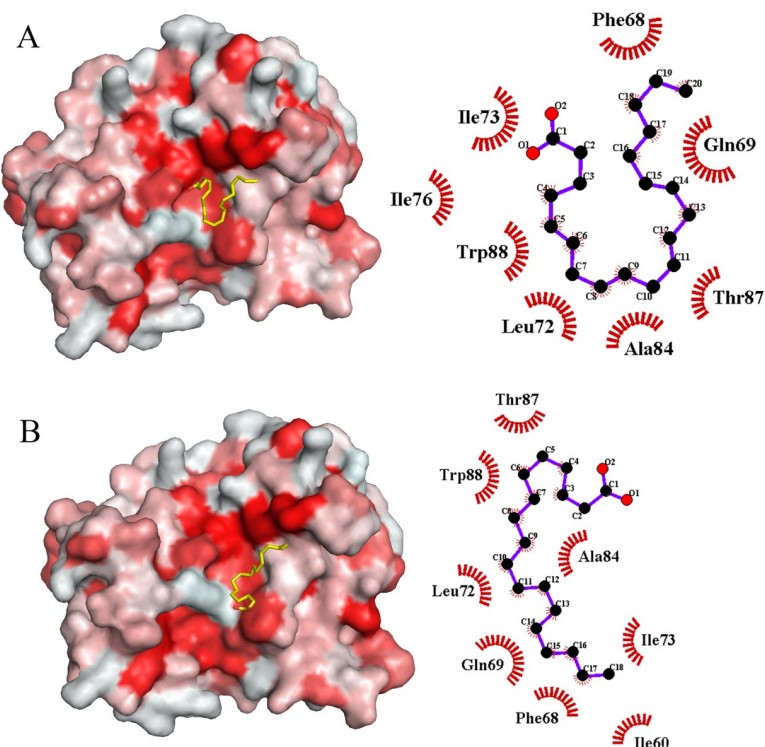

**Fig 3. The interaction of Calprotectin-fatty acids created by Lig-plot and Pymol software.** (A) The 3D and 2D images of Calprotectin-AA complex. (B) The 3D and 2D images of Calprotectin-OA complex. The 3D images indicate that the AA and OA bind to Hydrophobic regions between the two Calprotectin subunits (S100A8/S100A9) which were represented in red. The 2D images demonstrate the hydrophobic amino acids which involved in fatty acid binding.

Korndorfer et al confirmed that in binding two subunits to each other (A8/A9), the Phe68, Ile13, Ile12, Ile9, Leu72 and Gln69 amino acids of A8 subunit and the His91, Trp88 and Thr87 of A9 subunit are incorporated [21]. Furthermore, our results verified the binding of Phe68, Gln69 and Leu72 of A8 subunit and Thr87, Trp88 of A9 subunit which are the hydrophobic and joint region of two subunits to AA and OA. To evaluate the conformational changes of the second structure of Calprotectin-fatty acids during the MD simulation, the DSSP analysis was done [39]. As shown in the Fig 4, the βeta-sheet and helix content declined. The helix content declined from 36 ± 1% in Calprotectin alone to 35.5 ± 0.8% in Calprotectin-AA and 35 ± 1.4% in Calprotectin-OA complexes. Moreover, the βeta-sheet percentage in Calprotectin alone reduced from 25 ± 1.3% to 23.7 ± 0.7% in Calprotectin-AA and 20.6 ± 0.7% in Calprotectin-OA complex. Furthermore, other structures including turns and coils content in Calprotectin-AA/OA complexes increased in comparison with Calprotectin alone. Gheibi N et al, proved that presence of AA and OA in complex with Calprotectin cause helix and βeta-sheet content reduction in comparison with Calprotectin alone but the content of other structures increased in complexes in comparison with Calprotectin alone. Moreover, in the above study, the helix content in Calprotectin alone was 34.2%, while it was declined to 31.8% in Calprotectin-AA and 29.8% in Calprotectin-OA complexes. Furthermore, the β-sheet content declined from 23% in Calprotectin alone to 22.8% and 20.8% in Calprotectin-AA and Calprotectin-OA respectively. [20]. Thus, the experimental studies confirmed our results (Fig 4).

## 3.2. Analysis of the electrostatic and van der Waals energies

To evaluate the major interactions which involved in Calprotectin-AA/OA formation, the electrostatic and van der Waals energies were estimated between the aforementioned fatty acids and the Calprotectin, using MD simulation method [40]. van der Waals and electrostatic energies were illustrated in Fig 5.

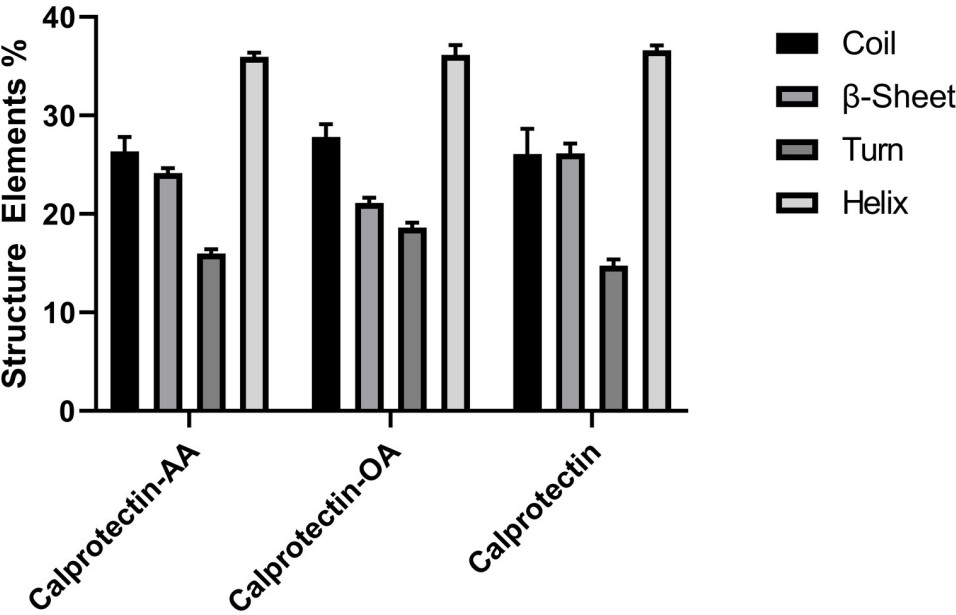

**Fig 4. The Secondary structure of Calprotectin alone, Calprotectin-AA and Calprotectin-OA using the DSSP algorithm.**

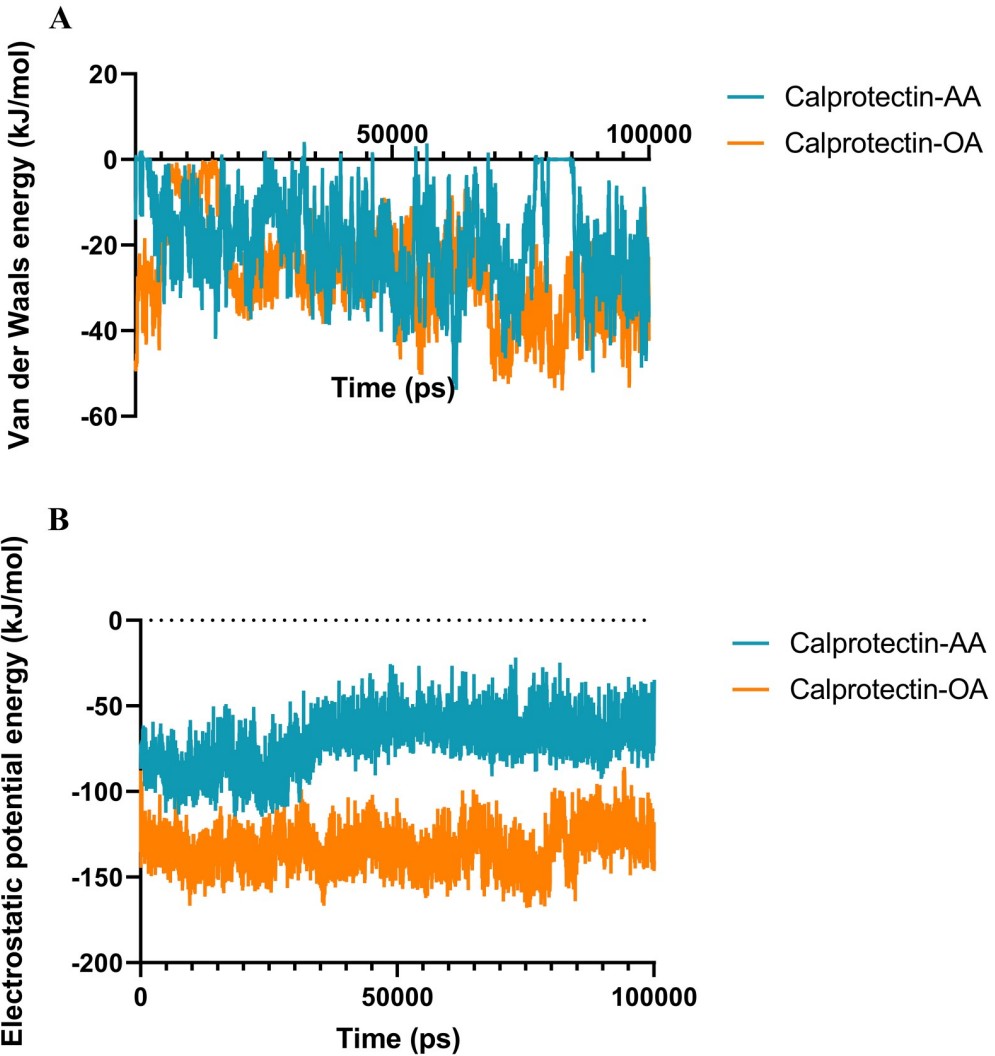

**Fig 5. The van der Waals and electrostatic energies between Calprotectin and fatty acids.** (A) The van der Waals energy of Calprotectin-AA and Calprotectin-OA complexes. (B) The electrostatic energy of Calprotectin-AA and Calprotectin-OA complexes. As shown, the van der Waals energies of the two complexes are almost similar but the electrostatic energy of the Calprotectin-OA complex is nearly 2-times greater than Calprotectin-AA.

The electrostatic energy contribution for the above complexes was significantly higher than van der Waals energy. The electrostatic energy of Calprotectin-AA and Calprotectin-OA complexes during the last 5 ns of simulation were -62.8 and -126.7 kJ/mol respectively. Moreover, the van der Waals energy of Calprotectin-AA and Calprotectin-OA complexes were -28.4 and -31.9 kJ/mol. These results demonstrated that the electrostatic and van der Waals energy difference between OA and Calprotectin was -63.9 and -3.5 kJ/mol higher than the Calprotectin-AA complex. Thus, these findings confirm that OA has a stronger interaction with Calprotectin in comparison with AA.

### 3.3. The binding energy of S100A8/A9 subunits in presence of AA/OA fatty acids

The MMPBSA is a method which is currently used to estimate the binding free energy [41]. Snapshots were extracted at every 50 ps of stable intervals from 80–100 ns MD trajectory. The

binding free energy of the complexes between S100A8 and S100A9 subunits was measured for Calprotectin alone, Calprotectin-AA and Calprotectin-OA using g_mmpbsa command in Gromacs software (S1 Table) [33]. The results revealed that Calprotectin possessed the highest negative binding free energy (-269.283 kJ/mol). Moreover, Calprotectin-AA and Calprotectin-OA complexes showed the affinity of -174.994 and -82.487 kJ/mol respectively. Thus, binding fatty acids (OA/AA) to Calprotectin led to instability of A8 and A9 subunits which confirmed by the above results. Furthermore, van der Waals and electrostatic interactions and non-polar solvation energy contributed negatively, while the polar solvation energy contributed positively to the total free binding energy. It should be noted that van der Waals interactions provide higher contribution than electrostatic interactions for all three complexes except the Calprotectin alone while the electrostatic interaction plays the main role in comparison with van der Waals in binding energy. Moreover, the non-polar free energy has the lowest contribution to the total binding energy (S1 Table).

### 3.4. Effects of AA and OA on H-bond interactions of the Calprotectin subunits

The formation of intermolecular H-bond in protein plays a great role in the stability of the system [42]. For a better recognition of the Calprotectin stability, the number of H-bonds between A8 and A9 subunits was calculated for Calprotectin alone and during the formation of Calprotectin-AA/Calprotectin-OA complexes. The Fig 6, shows the H-bond average between two subunits of Calprotectin in presence of AA and OA during the simulation process which accounted as 9 and 7 respectively, while the average number of H-bond for Calprotectin alone was 20 during the simulation. It seems that the AA and OA can induce the structural instability in Calprotectin heterodimer.

## Conclusion

Calprotectin is a major protein which involved in different abnormalities including inflammatory and cancer diseases. Moreover, the protein has the ability to bind to UFAs which is considered as a main carrier for fatty acids and also can cause UFAs movement between the

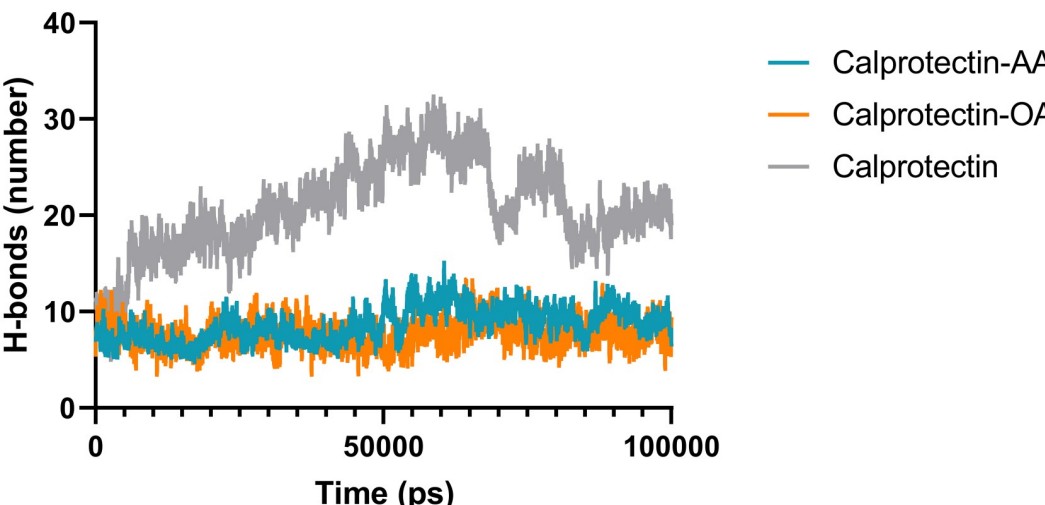

**Fig 6. The number of H-bonds between A8 and A9 subunits in Calprotectin-AA, Calprotectin-OA and the Calprotectin alone.** As illustrated, the H-bond content in the above complexes are lower than the Calprotectin alone.

cytosol of neutrophils and the plasma membrane which is essential for fatty acids uptake and the eicosanoids metabolism [43]. The MD results showed the structural changes and the protein flexibility of the Calprotectin in presence of fatty acids. Our findings revealed that the Calprotectin possesses the lowest flexibility in a majority of residues in presence of AA and OA which verifies the interaction of these fatty acids with Calprotectin. Moreover, the DSSP analysis was done to investigate the conformational changes of Calprotectin before and after binding to fatty acids. These structural changes were coincidence with β-sheet and helix content decline but adversely the turns and coils content were increased. In addition, the H-bond reduction in Calprotectin-AA/OA complexes confirms the instability of the heterodimer. Furthermore, the electrostatic energy measurements demonstrated a stronger interaction of OA with Calprotectin in comparison with AA. The binding free energy measurements indicated that the highest negative binding free energy belongs to the Calprotectin alone. Several studies confirmed that the active neutrophils which are dominant in inflammation, release the AA to plasma which can strengthen the active inflammatory response [44, 45]. These findings proved the specific binding sites of S100A8/A9 complex for AA in cytosol of the neutrophil which verified the essential role of the S100A8/A9 complex in metabolism of AA [46]. In one study, Eckert et al. confirmed that AA and OA may have a role in inflammation [47]. Based on these findings and due to the importance of Calprotectin protein in several human diseases and different cell processes and also due to importance of fatty acids in various cellular mechanisms, in this study we investigated the structural changes of Calprotectin alone and in complex with AA and OA, to provide a better understanding of the protein and its mechanism of action which could be useful in future drug design and treatment purposes.

## Supporting information

**S1 Table. MMPBSA free energies of Calprotectin-AA, Calprotectin-OA complexes and the Calprotectin alone.**
(DOCX)

## Acknowledgments

The authors appreciate the Research Council of Qazvin University of Medical Sciences.

## Author Contributions

**Conceptualization:** Mohamad Ghorbani, Alireza Farasat.

**Data curation:** Nematollah Gheibi.

**Formal analysis:** Mohamad Ghorbani.

**Investigation:** Hanifeh Shariatifar.

**Methodology:** Mohamad Ghorbani, Alireza Farasat.

**Project administration:** Alireza Farasat.

**Software:** Alireza Farasat.

**Supervision:** Alireza Farasat.

**Validation:** Nematollah Gheibi.

**Writing – original draft:** Hanifeh Shariatifar.

**Writing – review & editing:** Hanifeh Shariatifar.

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
