## [Decision Letter · Decision Letter 0]

28 Jan 2020

PONE-D-20-00531

Effects of unsaturated fatty acids (Arachidonic/Oleic Acids) on stability and structural properties of Calprotectin using Molecular Docking and Molecular Dynamics simulation approach

PLOS ONE

Dear Dr. Farasat,

Thank you for submitting your manuscript to PLOS ONE. After careful consideration, we feel that it has merit but does not fully meet PLOS ONE’s publication criteria as it currently stands. Therefore, we invite you to submit a revised version of the manuscript that addresses the points raised during the review process.

 Please try to improve your manuscript according to the reviewers' criticism.

We would appreciate receiving your revised manuscript by Mar 13 2020 11:59PM. To enhance the reproducibility of your results, we recommend that if applicable you deposit your laboratory protocols in protocols.io, where a protocol can be assigned its own identifier (DOI) such that it can be cited independently in the future. For instructions see: http://journals.plos.org/plosone/s/submission-guidelines#loc-laboratory-protocols

We look forward to receiving your revised manuscript.

Kind regards,

Eugene A. Permyakov, Ph.D., Dr.Sci.

Academic Editor

PLOS ONE

Journal Requirements:

Reviewers' comments:

Reviewer's Responses to Questions

**Comments to the Author**

1. Is the manuscript technically sound, and do the data support the conclusions?

Reviewer #1: Yes

Reviewer #2: Partly

2. Has the statistical analysis been performed appropriately and rigorously? 

Reviewer #1: Yes

Reviewer #2: No

3. Have the authors made all data underlying the findings in their manuscript fully available?

Reviewer #1: Yes

Reviewer #2: Yes

4. Is the manuscript presented in an intelligible fashion and written in standard English?

Reviewer #1: Yes

Reviewer #2: Yes

5. Review Comments to the Author

Reviewer #1: The manuscript "Effects of unsaturated fatty acids (Arachidonic/Oleic Acids) on stability and structural properties of Calprotectin using Molecular Docking and Molecular Dynamics simulation approach" by Alireza Farasat et al. described the effects of fatty acids on Calprotectin.

I recommend the manuscript for publication in PLOS ONE after making minor changes in the text.

1. The Authors should avoid acronyms in the abstract.

2. My main remarks concern the text in manuscript to Figure 3 on pages 9 and 10. Phe68 and Thr87 are not present in Figure 3A. Glu69 must be replaced by Gln69 in all places. ILe60 is not written correctly, change to Ile60. Phe68 and Thr87 are not common. IL72 must be replaced with Leu72.

3. Further comments on the text.

page 10. "As shown in the figure, the beta-sheet..." What figure?

page 10. "Previous studies proved...." Reference is absent.

Reviewer #2: The authors present a paper in which they study interactions of fatty acids with calprotein. The authors study an important topic and their computational approach is appropriate. However, the presented data on structural changes does not support the conclusions. Specifically, the presented values in declined beta-sheet and helix content are very similar and no error estimates for these values are given. The authors should repeat the simulations of apo calprotein , and with AA/OA complexes and calculate standard errors based on the results of the two independent simulations. This is always a good practice in molecular dynamics simulations. Additionally, comparison to the results of experimental structural content in reference 20 should be done. I recommend the publication of this paper after additional simulations and statistical analysis, as well correction of the text below.

1, Correct capitalization of "van der Waals" and "electrostatic" should be used.

2. the beta symbol should be used instead of "beta" in beta-sheets.

3. On page 4 sentence “Moreover, the Calprotectin has the ability

to bind UFAs comprising AA, Linoleic acid (LA) and Oleic acid (OA) in a calcium dependent

manner which is considered as the main fatty acid carrier of neutrophil with high binding affinity

potential.” is a bit weird, I propose to replace it with

“The calprotectin is considered to be he main fatty acid carrier of neutrophils with high binding affinity

potential and can bind UFAs such as AA, Linoleic acid (LA) and Oleic acid (OA) in a calcium dependent manner. “

4. Also in the next sentence on page 4 , “The following attachment” should be replaced with “This binding” for clarity.

5. Page 6 end of introduction “(TLR4, RAGE, ....)” replace "… " with something else or remove

6. In CaCl2 2 should be subscripted, as in all chemical formulas.

7. "H-bond" abbreviation should be spelled out when it is introduced and added to the list of abbreviations.

8. On page 10 in sentence “As shown in the figure, the beta-sheet and helix content declined.” the figure number is missing. Also right after this sentence, "Such a way” does not fit with the text, I propose to remove it.

9. Also on page 10 the sentence "The percentage of turn structure was increased from 14% to 18% in Calprotectin-OA complex, the percentage of coils in Calprotectin was 25% while, in Calprotectin-OA complex has been reached to 26.4%" this sentence should be rewritten for clarity, please specify what systems are compared for values 14% and 18% .

10. Reference 31 is missing journal name volume and pages

11. The resolution of the figures should be improved before publication.

6. PLOS authors have the option to publish the peer review history of their article (what does this mean?). If published, this will include your full peer review and any attached files.

Reviewer #1: No

Reviewer #2: No

---

## [Author Response · Author response to Decision Letter 0]

8 Mar 2020

Reviewer 1: 

The manuscript "Effects of unsaturated fatty acids (Arachidonic/Oleic Acids) on stability and structural properties of Calprotectin using Molecular Docking and Molecular Dynamics simulation approach" by Alireza Farasat et al. described the effects of fatty acids on Calprotectin.

I recommend the manuscript for publication in PLOS ONE after making minor changes in the text.

1. The Authors should avoid acronyms in the abstract.

Reply: In the abstract part, the acronyms were omitted.

2. My main remarks concern the text in manuscript to Figure 3 on pages 9 and 10. Phe68 and Thr87 are not present in Figure 3A. Glu69 must be replaced by Gln69 in all places. ILe60 is not written correctly, change to Ile60. Phe68 and Thr87 are not common. IL72 must be replaced with Leu72.

Reply: The above changes were done.

3. Further comments on the text.

page 10. "As shown in the figure, the beta-sheet..." What figure?

page 10. "Previous studies proved...." Reference is absent.

Reply: The aforementioned changes were done.

Reviewer 2: The authors present a paper in which they study interactions of fatty acids with calprotein. The authors study an important topic and their computational approach is appropriate. However, the presented data on structural changes does not support the conclusions. Specifically, the presented values in declined beta-sheet and helix content are very similar and no error estimates for these values are given. The authors should repeat the simulations of apo calprotein and with AA/OA complexes and calculate standard errors based on the results of the two independent simulations. This is always a good practice in molecular dynamics simulations. Additionally, comparison to the results of experimental structural content in reference 20 should be done. I recommend the publication of this paper after additional simulations and statistical analysis, as well correction of the text below.

1. Correct capitalization of "van der Waals" and "electrostatic" should be used.

Reply: Capitalization was corrected.

2. the beta symbol should be used instead of "beta" in beta-sheets.

Reply: The β symbol was applied to the manuscript.

3. On page 4 sentence “Moreover, the Calprotectin has the ability

to bind UFAs comprising AA, Linoleic acid (LA) and Oleic acid (OA) in a calcium dependent

manner which is considered as the main fatty acid carrier of neutrophil with high binding affinity

potential.” is a bit weird, I propose to replace it with

“The calprotectin is considered to be the main fatty acid carrier of neutrophils with high binding affinity

potential and can bind UFAs such as AA, Linoleic acid (LA) and Oleic acid (OA) in a calcium dependent manner. “

Reply: On page 4, the sentence was replaced with your desired phrase.

4. Also in the next sentence on page 4, “The following attachment” should be replaced with “This binding” for clarity.

Reply: It was replaced with “This binding” phrase.

5. Page 6 end of introduction “(TLR4, RAGE, ....)” replace "… " with something else or remove

Reply: It was removed.

6. In CaCl2 2 should be subscripted, as in all chemical formulas.

Reply: It was done.

7. "H-bond" abbreviation should be spelled out when it is introduced and added to the list of abbreviations.

Reply: It was done.

8. On page 10 in sentence “As shown in the figure, the beta-sheet and helix content declined.” the figure number is missing. Also right after this sentence, "Such a way” does not fit with the text, I propose to remove it.

Reply: It was removed based on your valuable comment.

9. Also on page 10 the sentence "The percentage of turn structure was increased from 14% to 18% in Calprotectin-OA complex, the percentage of coils in Calprotectin was 25% while, in Calprotectin-OA complex has been reached to 26.4%" this sentence should be rewritten for clarity, please specify what systems are compared for values 14% and 18%.

Reply: Changes were done.

10. Reference 31 is missing journal name volume and pages.

Reply: As we checked the google scholar, unfortunately the aforementioned article, doesn’t have volume no., but the pages were added.

11. The resolution of the figures should be improved before publication.

Reply: The resolution of the figures was improved.

---

## [Editor Report · Decision Letter 1]

10 Mar 2020

Effects of unsaturated fatty acids (Arachidonic/Oleic Acids) on stability and structural properties of Calprotectin using Molecular Docking and Molecular Dynamics simulation approach

PONE-D-20-00531R1

Dear Dr. Farasat,

We are pleased to inform you that your manuscript has been judged scientifically suitable for publication and will be formally accepted for publication once it complies with all outstanding technical requirements.

With kind regards,

Eugene A. Permyakov, Ph.D., Dr.Sci.

Academic Editor

PLOS ONE
---

## [Editor Report · Acceptance letter]

12 Mar 2020

PONE-D-20-00531R1 

Effects of unsaturated fatty acids (Arachidonic/Oleic Acids) on stability and structural properties of Calprotectin using Molecular Docking and Molecular Dynamics simulation approach 

Dear Dr. Farasat:

I am pleased to inform you that your manuscript has been deemed suitable for publication in PLOS ONE. Congratulations! Your manuscript is now with our production department. 

With kind regards,

on behalf of

Prof. Eugene A. Permyakov 

Academic Editor

PLOS ONE